# A Genome-Wide Association Study of Metabolic Syndrome in the Taiwanese Population

**DOI:** 10.3390/nu16010077

**Published:** 2023-12-25

**Authors:** Chih-Yi Ho, Jia-In Lee, Shu-Pin Huang, Szu-Chia Chen, Jiun-Hung Geng

**Affiliations:** 1School of Post-Baccalaureate Medicine, College of Medicine, Kaohsiung Medical University, Kaohsiung 807, Taiwan; u109000017@gap.kmu.edu.tw; 2Department of Psychiatry, Kaohsiung Medical University Hospital, Kaohsiung Medical University, Kaohsiung 807, Taiwan; 1050644@kmuh.org.tw; 3Department of Urology, Kaohsiung Medical University Hospital, Kaohsiung Medical University, Kaohsiung 807, Taiwan; 860102@kmuh.org.tw; 4Department of Urology, School of Medicine, College of Medicine, Kaohsiung Medical University, Kaohsiung 807, Taiwan; 5Graduate Institute of Clinical Medicine, College of Medicine, Kaohsiung Medical University, Kaohsiung 807, Taiwan; 6Research Center for Environmental Medicine, Kaohsiung Medical University, Kaohsiung 807, Taiwan; 920497@kmuh.org.tw; 7Ph.D. Program in Environmental and Occupational Medicine, College of Medicine, Kaohsiung Medical University, Kaohsiung 807, Taiwan; 8Institute of Medical Science and Technology, College of Medicine, National Sun Yat-Sen University, Kaohsiung 804, Taiwan; 9Department of Internal Medicine, Kaohsiung Municipal Siaogang Hospital, Kaohsiung Medical University, Kaohsiung 812, Taiwan; 10Division of Nephrology, Department of Internal Medicine, Kaohsiung Medical University Hospital, Kaohsiung Medical University, Kaohsiung 807, Taiwan; 11Faculty of Medicine, College of Medicine, Kaohsiung Medical University, Kaohsiung 807, Taiwan; 12Department of Urology, Kaohsiung Municipal Siaogang Hospital, Kaohsiung 812, Taiwan

**Keywords:** GWAS, metabolic syndrome, Taiwan biobank, single nucleotide polymorphism, hypertension, diabetes mellitus, waist circumference, triglyceride, high-density lipoprotein cholesterol

## Abstract

The purpose of this study was to investigate genetic factors associated with metabolic syndrome (MetS) by conducting a large-scale genome-wide association study (GWAS) in Taiwan, addressing the limited data on Asian populations compared to Western populations. Using data from the Taiwan Biobank, comprehensive clinical and genetic information from 107,230 Taiwanese individuals was analyzed. Genotyping data from the TWB1.0 and TWB2.0 chips, including over 650,000 single nucleotide polymorphisms (SNPs), were utilized. Genotype imputation using the 1000 Genomes Project was performed, resulting in more than 9 million SNPs. MetS was defined based on a modified version of the Adult Treatment Panel III criteria. Among all participants (mean age: 50 years), 23% met the MetS definition. GWAS analysis identified 549 SNPs significantly associated with MetS, collectively mapping to 10 genomic risk loci. Notable risk loci included rs1004558, rs3812316, rs326, rs4486200, rs2954038, rs10830963, rs662799, rs62033400, rs183130, and rs34342646. Gene-set analysis revealed 22 associated genes: *CETP*, *LPL*, *APOA5*, *SIK3*, *ZPR1*, *APOC1*, *BUD13*, *MLXIPL*, *TOMM40*, *GCK*, *YKT6*, *RPS6KB1*, *FTO*, *VMP1*, *TUBD1*, *BCL7B*, *C19orf80 (ANGPTL8)*, *SIDT2*, *SENP7*, *PAFAH1B2*, *DOCK6*, and *FOXA2.* This study identified genomic risk loci for MetS in a large Taiwanese population through a comprehensive GWAS approach. These associations provide novel insights into the genetic basis of MetS and hold promise for the potential discovery of clinical biomarkers.

## 1. Introduction

Metabolic syndrome (MetS) encompasses a constellation of metabolic risk factors, including central obesity, hypertension, insulin resistance, and dyslipidemia, all of which are interlinked with atherosclerosis [1]. MetS is notably associated with an elevated risk of type 2 diabetes, fatty liver disease, chronic kidney disease, cardiovascular disease, and cancer [2,3,4]. While the precise prevalence of MetS varies according to different definitions, its global incidence has been steadily increasing. A systematic review in 2017 revealed that a substantial proportion, exceeding one-fifth of the adult population in the Asia-Pacific region, has MetS [5]. Drawing from NHANES data spanning 1988 to 2010 in the United States, there has been an average annual increase of 0.37% in body mass index (BMI) for both genders, accompanied by a corresponding increase of 0.27% per year in waist circumference among women. According to data from the Centers for Disease Control and Prevention (CDC) in 2017, an estimated 30.2 million adults in the United States, constituting 12.2% of the population, have type 2 diabetes, while approximately one-third of adults have MetS [6]. In Taiwan, there has been a notable increase in the prevalence of MetS, affecting 9.5% of the population, based on the strict NCEP III standards. In addition, the prevalence rises to 12.9% when considering the Asian waist circumference criteria [7]. MetS poses a formidable global health challenge, shaped by an intricate interplay of multiple factors. Despite extensive research, the fundamental genetic underpinnings of MetS have yet to be fully elucidated.

The genetic factors contributing to MetS encompass intricate interactions among multiple genes and environmental influences [8]. While lifestyle choices, notably diet and exercise, wield significant impact, genetic predispositions play a substantial role in determining an individual’s susceptibility to this condition [9]. Several studies have identified numerous genes linked to lipid metabolism, insulin resistance, inflammation, and obesity, with specific variants such as *LPL*, *CETP*, and *APOA5* demonstrating heightened susceptibility [10,11,12]. The heritability of MetS, observable in families with a history of diabetes, hypertension, or obesity, underscores its genetic foundation [13]. Understanding the genetic basis of MetS is critical for tailored medical approaches, yet the condition’s polygenic nature complicates the identification of individual genes. Large-scale studies aimed at identifying genetic markers offer promise in unraveling its complexity.

Genome-wide association studies (GWASs) have been developed over the past two decades and have emerged as a potent approach for investigating the genetic underpinnings of complex diseases across diverse disciplines. By systematically scrutinizing individual genomes, GWASs enable the exploration of genetic variations within the genome to discern associations between genotypes and phenotypes. This technique has brought about a revolutionary shift in the once intricate and elusive realm of genetics, yielding robust evidence for the connections between numerous diseases and genetic variations [14]. At present, the results of GWASs have confirmed associations between genetic variations and a multitude of diseases, including coronary artery disease, type 2 diabetes, nephrolithiasis, and specific cancers [15,16,17,18]. As such, applying a GWAS approach within the context of MetS holds the potential to reveal the genetic factors that contribute to its underlying causes.

While several large-scale studies have applied a GWAS approach to MetS, most of the identified genetic loci have originated from research conducted in Western countries. While recent investigations have begun to focus on Asian populations, including the Han Chinese population, these efforts remain comparatively limited. Therefore, the aim of this study was to conduct an expansive GWAS tailored specifically to the Taiwanese population, with the goal of identifying single nucleotide polymorphisms (SNPs) linked with MetS in this population, and consequently address the existing scarcity of data concerning Asian populations in contrast to their Western counterparts.

## 2. Materials and Methods

### 2.1. Taiwan Biobank and Study Population

This study sourced its data from the Taiwan Biobank (TWB), an extensive community research database established in 2005 to serve as a pivotal biomedical research asset [19]. The TWB’s principal objective revolves around creating a repository of biological samples and pertinent health-related information, thereby enabling the ongoing surveillance of Taiwan’s population. This invaluable resource facilitates the exploration of genetics, environmental influences, lifestyle factors, and their intricate connections with various diseases and health outcomes. The TWB undertook the recruitment of a diverse spectrum of participants, ranging in age from 30 to 70 years, from across Taiwan through structured questionnaires, interviews, and comprehensive clinical assessments. Employing this comprehensive approach, detailed demographic information, health statuses, and lifestyle factors were meticulously documented [20,21,22,23,24]. A grand total of 107,230 participants, each possessing ample baseline information and successfully meeting the quality control criteria for a GWAS, were recruited for inclusion in this study. Prior to their inclusion, informed consent was obtained from all participants, and all data-sharing protocols and policies were meticulously followed. The Institutional Review Board of Kaohsiung Medical University Hospital provided ethical oversight for the study (Approval No. KMUHIRB-E(I)-20210212).

### 2.2. Phenotypic Data

To identify individuals with MetS, we used the diagnostic criteria outlined by the Taiwan Ministry of Health Services [25]. The presence of three or more of the following five risk factors was used to identify MetS: (1) waist circumference of ≥90 cm for men and ≥80 cm for women, (2) systolic blood pressure > 130 mm Hg or diastolic blood pressure > 85 mm Hg, (3) high-density lipoprotein (HDL) cholesterol level < 40 mg/dL for males, <50 mg/dL for females, (4) fasting blood glucose level ≥ 100 mg/dL, and (5) triglyceride level ≥ 150 mg/dL. The phenotypic data necessary to define MetS were collected during the participants’ evaluations.

### 2.3. Genotyping

The Whole Genome Genotyping Array from Affymetrix (Santa Clara, CA, USA) was used to conduct genotypic analysis on participants in the TWB [19]. To ensure the meticulous selection of specific SNPs suitable for genetic trait analysis within the Taiwanese Han Chinese population, the TWB adopted Affymetrix Power Tools as a standardized quality-control protocol. SNPs situated on both the X and Y chromosomes, as well as those present in mitochondrial DNA, were included in the released dataset. This included the Affymetrix TWB 1.0 SNP chip (comprising around 650,000 SNPs) and TWB 2.0 SNP chip (comprising around 750,000 SNPs), which was utilized for 26,168 and 81,026 participants in the present study, respectively. The TWB 1.0 chips, developed in collaboration with Thermo Fisher Scientific in the United States (Waltham, MA, USA) and the National Genome Research Center (National Center for Genome Medicine, NCGM) in Taiwan, incorporate cutting-edge technology. Together, they have designed an exclusive gene-typing chip for the Taiwanese Han population, focusing on 653,291 carefully selected SNPs. These SNPs were sourced from diverse origins: (1) Axiom genome-wide CHB (Han Chinese in Beijing, China) array: This array contributed a total of 525,652 SNPs. (2) Published cancer genome-wide association studies: Statistically significant SNPs identified in these studies were included. (3) NCGM’s previous research results: Samples from past studies involving various chips were analyzed, specifically focusing on Chinese individuals, to identify relevant SNPs. (4) Whole-exome sequencing and other sequencing methods: SNPs displaying polymorphism in Chinese samples were meticulously chosen through comprehensive sequencing research methods. (5) Other SNPs related to drug response and metabolism: Genes like MHC and PGX, associated with drug response and metabolism, were explored for additional relevant SNPs. The development of TWB 2.0 chips followed a similar approach to TWB 1.0 chips, and comprehensive details about the distinctions between TWB 1.0 and 2.0 SNP chips can be accessed via the official website (https://www.twbiobank.org.tw, accessed on 7 November 2023).

### 2.4. Quality Control

An analysis of the Affymetrix microarray data and quality control enforcement throughout the procedure was carried out using PLINK 1.9 [26], which also incorporated the Hardy–Weinberg equilibrium (HWE) test. Appendix A presents the flowchart, and detailed criteria are described below. Samples and SNPs not meeting any of the following criteria were excluded from the analysis: (1) ambiguous sex data, (2) call rate below 95%, (3) samples displaying heterozygosity levels deviating beyond ± 3 SD from the mean, (4) significant deviation from the HWE (*p* < 1 × 10^−5^), (5) relatedness (PI-hat > 0.1875), and (6) a minor allele frequency (MAF) < 0.05 [27].

### 2.5. Imputation

We conducted genotype imputation using SHAPEIT (v2.r790) [28] and IMPUTE2 (v2.3.1) [29]. The reference panel consisted of whole-genome sequencing data from a Taiwanese population and an East Asian population from the 1000 Genomes Project. Subsequent to imputation, we excluded multiallelic sites and SNPs failing post-imputation criteria: call rate > 0.05, MAF < 0.01, or R^2^ < 0.3. This filtering yielded a final set of 9,809,486 variants prepared for subsequent association analysis. Detailed information has been described earlier [19].

### 2.6. Association Analysis

We performed SNP-based association analysis using PLINK (v1.9) [26] and logistic regression. Prior to initiating the analysis, we adjusted for age, sex, and the top 10 principal components of ancestry during phenotype preparation. The multivariate logistic regression models for GWAS are represented as follows: Y = β_0_ + β_1_X_1_ + β_2_X_2_ + … + β_n_X_n_, where Y represents the dependent variable (MetS), taking the value Y = 1 when MetS occurs and Y = 0 when it does not… X_1_, X_2_, …; X_n_ denote the predictors (genetic variants or other variables); and β_0_, β_1_, β_2_, …, β_n_ represent the beta coefficients associated with these predictors. Within our study, GWAS results encompassed SNP names, chromosomes, positions, effect alleles, other alleles, effect allele frequencies, beta coefficients, standard errors of the beta coefficients, *p*-values, and nearest genes. The beta coefficient reflects the effect size of a genetic variant on the MetS trait. A positive beta coefficient suggests that an increase in the specific allele aligns with an increase in the trait, whereas negative values indicate a decrease. These coefficients, combined with their respective standard errors and *p*-values, contribute to assessing the significance and direction of genetic variants’ influence on MetS risk. To visualize the results, we generated Manhattan plots and quantile–quantile (Q–Q) plots using the R CMplot package (v3.6.2, R Foundation for Statistical Computing, Vienna, Austria).

In this study, we defined independent significant SNPs as those with a *p*-value ≤ 5 × 10^−8^ that were independent of each other at r^2^ < 0.6. Genomic risk loci were subsequently identified as specific regions encompassing clusters of independent signals. These loci were demarcated by the presence of independent significant SNPs sharing dependencies with an r^2^ ≥ 0.1 while simultaneously being situated within a designated distance of 250 kb. For the genetic risk loci analysis, we used Functional Mapping and Annotation of GWAS (FUMA) v1.3.4 [30,31].

### 2.7. Functional Annotation and Pathway-Enrichment Analyses

To comprehensively investigate the underlying genetic factors of MetS through association analysis, we employed a comprehensive suite of bioinformatics methodologies. Initially, we used the Multi-marker Efficient Mixed Model Association (MEGMA) tool [32] to annotate significant SNPs identified from the GWAS. This involved mapping the genomic coordinates of these SNPs to neighboring genes, facilitating the establishment of robust SNP-to-gene associations. We used the genome assembly GRCh38 (NCBI 38) [33] datasets and tested a total of 15,290 genes. Based on Bonferroni correction, significant genes were defined as those with a *p*-value ≤ 3.27 × 10^−6^. To visualize the results, we used the gene-based test computed by MAGMA based on a GWAS for MetS summary statistics using the R CMplot package (v3.6.2, R Foundation for Statistical Computing, Vienna, Austria). Following this functional annotation, we conducted pathway-enrichment analyses using curated gene sets (C2) of Kyoto Encyclopedia of Genes and Genomes (KEGG) datasets in MSigDB (Kanehisa Laboratories, Kyoto, Japan) [34] to shed light on the underlying biological pathways in which these genes might be involved.

### 2.8. Statistical Analysis

We used a case-control study design for the GWAS of MetS. Cases were individuals meeting the previously described diagnostic criteria for MetS, while controls were individuals without MetS. The clinical characteristics of subjects, represented as mean ± standard deviation (SD) for continuous variables and as n (%) for categorical variables, underwent comparison using the chi-square test for categories and the independent *t*-test for continuous traits, considering a *p*-value < 0.05 as indicative of significant differences. We conducted a multivariate logistic regression analysis, adjusting for age, sex, and the top 10 principal components. We applied a genome-wide significance threshold of *p*-value ≤ 5 × 10^−8^ to identify genetic variants significantly associated with MetS.

## 3. Results

### 3.1. Clinical Characteristics of the Study Participants

A total of 107,230 participants from the TWB were included in this study, offering a comprehensive overview of the health status within the Taiwanese population. The collected data, as summarized in Table 1, provide essential insights into the clinical characteristics of the study participants. The average age of the participants was approximately 50 years, and the cohort was predominantly composed of females, accounting for 64% of the total participants. BMI measurements revealed an average of 24 kg/m2, indicating a generally healthy weight distribution among the participants. In terms of blood pressure, the mean systolic and diastolic values were 120 mmHg and 74 mmHg, respectively, falling within the normal range. Upon analyzing the participants for MetS and its components, it was found that 23% of the individuals met the MetS criteria. The prevalence rates of hypertension, impaired glucose tolerance, increased waist circumference, hypertriglyceridemia, and low HDL cholesterol were 35%, 21%, 47%, 21%, and 26%, respectively. A comparison between subjects with and without MetS showed that those with MetS were older, had a higher BMI, and exhibited elevated blood pressure levels (Table 1).

### 3.2. Genomic Risk Loci for MetS

We performed a comprehensive GWAS analysis on our cohort, leading to the discovery of 549 SNPs significantly associated with MetS (Appendix A). The Manhattan plot and Q–Q plot of the MetS GWAS, depicted in Figure 1, highlighted significant loci clustering on chromosomes 7, 8, 11, 16, and 19. The Q–Q plot resulting from our GWAS effectively contrasts the observed distribution of *p*-values for genetic variants across the genome against their expected values. Our analysis highlights a significant deviation from the expected null distribution, indicating an increased occurrence of lower *p*-values beyond chance alone. Notably, our analysis identified ten key SNPs linked to MetS, each with distinct genetic variations and implications. These SNPs, namely rs1004558, rs3812316, rs326, rs4486200, rs2954038, rs10830963, rs662799, rs62033400, rs183130, and rs34342646, are detailed in Table 2. Each SNP is characterized by specific chromosomal location, effect allele, frequency, and association statistics, providing valuable insights into the genetic underpinnings of MetS within our study population.

### 3.3. Gene-Based Analysis for Mets

To evaluate the statistical enrichment and provide functional annotations and analysis of specific gene sets related to MetS in this study, we used MEGMA. Table 3 showcases 22 key genes identified in our analysis, each potentially contributing to the genetic basis of MetS in our study population. These genes include CETP, LPL, APOA5, SIK3, ZPR1, APOC1, BUD13, MLXIPL, TOMM40, GCK, YKT6, RPS6KB1, FTO, VMP1, TUBD1, BCL7B, C19orf80 (ANGPTL8), SIDT2, SENP7, PAFAH1B2, DOCK6, and FOXA2. The Manhattan plot presented in Figure 2 visually represents the genetic information of these genes, providing a comprehensive overview of their significance in the context of MetS.

### 3.4. Pathway Analysis for Mets

Finally, we used KEGG pathway analysis to understand the biological functions and interplay of these genes and their involvement in signaling pathways. The outcomes of this analysis are depicted in a visually informative form in Figure 3. Through this analysis, we gained insights into the pathways shaping the development and progression of MetS in our study cohort. Among the myriad pathways scrutinized, glycerolipid metabolism, the PPAR (Peroxisome Proliferator-Activated Receptor) signaling pathway, and fatty acid metabolism emerged as the three most significant signaling pathways influencing MetS within our participant pool.

## 4. Discussion

In this study, we used data from the TWB to determine the composition of MetS and its associated specific SNPs and genomic regions in the Taiwanese population. We found that approximately 20% of the included participants had MetS. Further GWAS analysis revealed that 10 SNPs (rs1004558, rs3812316, rs326, rs4486200, rs2954038, rs10830963, rs662799, rs62033400, rs183130, and rs34342646) were significantly associated with MetS, primarily located on chromosomes 7, 8, 11, 16, and 19. In addition, through MEGMA analysis, we identified 22 key genes (*CETP*, *LPL*, *APOA5*, *SIK3*, *ZPR1*, *APOC1*, *BUD13*, *MLXIPL*, *TOMM40*, *GCK*, *YKT6*, *RPS6KB1*, *FTO*, *VMP1*, *TUBD1*, *BCL7B*, *C19orf80 (ANGPTL8)*, *SIDT2*, *SENP7*, *PAFAH1B2*, *DOCK6*, and *FOXA2*) that were associated with MetS. Subsequent KEGG pathway analysis revealed that the three most significant signaling pathways related to MetS were glycerolipid metabolism, the PPAR signaling pathway, and fatty acid metabolism. In summary, this study represents the most comprehensive GWAS analysis of MetS in an Asian population to date. We identified novel associations between MetS and genes such as *SIK3*, *YKT6*, *RPS6KB1*, and *SENP7*, which have been less frequently reported in previous studies. Additionally, we conducted a functional analysis of these genes and identified their potential impact on molecular pathways, establishing a significant foundation for genetic analysis of MetS in Asian populations.

Comparing our findings with other studies reveals a substantial overlap of gene associations with MetS, emphasizing their relevance across populations and contexts [35,36,37,38,39,40,41,42,43,44,45]. Notably, the genes *LPL*, *CETP*, *APOA5*, *BUD13*, *ZPR1* and *GCK* have been shown to be consistently significantly associated with MetS across multiple studies [35,36,37,38,39,40,41,42,43,44,45]. *LPL* plays a role in lipoprotein metabolism and fatty acid utilization. It hydrolyzes triglycerides contained in circulating lipoproteins. Alterations or dysfunction of *LPL* can lead to disturbances in lipid processing, potentially contributing to MetS [46]. Similarly, *CETP* transfers cholesterol esters and triglycerides between lipoproteins, significantly impacting cholesterol metabolism and lipid balance [35,36,37]. These roles in MetS are supported by previous studies, including a cross-sectional analysis involving Mexican–Caucasian women [36]. *APOA5* plays a crucial role in lipid metabolism [38], and its mutations have been linked to abnormal lipid levels and increased cardiovascular risks [38]. Several studies have highlighted the role of *APOA5* in MetS and lipid-related issues [39,40]. Genetic variations in *BUD13*, *ZPR1*, and *APOA5* have been implicated in MetS-related events, showing significant correlations with elevated serum triglyceride levels and reduced serum HDL cholesterol [41,42]. *GCK* plays a crucial role in glucose phosphorylation, and it has been linked with diabetes and metabolic disorders [43,44,45]. The similar associations between these genes and MetS in our study and others shed light on the key factors in MetS across various genetic and contextual backgrounds.

The *SIK3*, *YKT6*, *RPS6KB1*, and *SENP7* genes have received comparatively less attention in previous research. *SIK3* encodes a protein involved in regulating glucose and lipid metabolism, as well as neuronal development [47,48]. It holds the potential to function as a candidate gene for obesity, metabolic disorders, and neurodegenerative diseases [47,48]. *YKT6*, a member of the SNARE protein family, has been associated with CD8+ T cell levels and is being considered as a potential biomarker for oral squamous cell carcinoma. Although it has been explored primarily in the context of cancer, its relevance to MetS has prompted further investigation [49,50,51]. *RPS6KB1*, a kinase linked to various cellular processes and protein synthesis, has demonstrated associations with conditions such as fatty liver disease, intestinal disorders, and oxidative stress pathways [52,53]. Meanwhile, *SENP7*, a regulator of SUMOylation, has been shown to play a role in maintaining the metabolic status of CD8+ T cells and contribute to muscle sarcomere organization [54,55]. The significance of these genes in relation to MetS underscores the imperative for further research into their mechanisms and to explore potential therapeutic applications.

Lind et al. conducted a GWAS of MetS using data of 291,107 individuals from the UK Biobank and identified 93 loci significantly linked to MetS. They also identified novel genes, including *WDR48*, *KLF14*, *NAADL1*, *GADD45G*, and *OR5R1* [56]. Their unique MetS-associated loci were not evident in our results, suggesting that different genes impact disease across different populations, although further studies are needed to validate this issue. A study conducted in the Mediterranean area in 2020 found that the *FADS1* gene cluster influenced the levels of omega-3 polyunsaturated fatty acids (PUFAs), and in particular that the *FADS1*-rs174547 T > C variant was strongly correlated [57]. Although our focus was also MetS, we did not find a significant link between *FADS1* and MetS, potentially due to their use of serum omega-3 PUFA versus our MetS-centered analysis, indicating the possibility of distinct results from refined metabolic pathways and interactions. Prasad et al. analyzed 10,093 Indo-European individuals and found robust correlations between *CETP* and *SFRP1* with MetS [35]. While *CETP* was identified in both their and our studies, their *SFRP1* investigation involved DNA methylation at rs16890462, affecting adipose-related functions, and this SNP was not found in our analysis. A Mexican study of 411 participants conducted in 2020 revealed correlations between MetS with rs1784042 and rs17120425 [36]. However, these SNPs were not relevant to MetS in our findings. In 2021, the GENNID study explored MetS genetics across diverse ethnicities and identified quantitative trait nucleotides for eight MetS traits, including the specific effect of rs186742063 in European Americans [58]. This SNP was not significantly associated with MetS in our analysis. Their multi-ethnic approach differs from our Taiwanese focus and highlights the importance of comprehensive population studies.

Most MetS research has focused on European and American populations; however, some studies have also been conducted in Asia [47,59,60]. In a 2019 GWAS study of 9676 Korean individuals with an average age of about 50 years, approximately 20% had MetS, which is consistent with our findings [60]. The authors highlighted that the *APOA5* genetic variant rs662799 was significantly associated with MetS, which is also consistent with our findings. Another GWAS study by Kong et al. of 9932 Korean women identified 14 distinct SNPs (rs2209363, rs768072, rs284544, rs284541, rs2012243, rs10947646, rs2283113, rs16923249, rs9568558, rs9516416, rs6492111, rs4072617, rs8107274, rs2827976) associated with MetS [47]. Their study, restricted to females, found differences in these 14 SNPs compared to our study, suggesting sex-based heterogeneity beyond ethnic factors. In a 2022 study conducted in Taiwan, which examined the relationship between the *APOE* locus and MetS, the presence of *APOE* ε2, ε3, and ε4 variants (identified through genetic markers rs7412 and rs429358) was associated with MetS [60]. Of particular note, the variant rs429358 was independently correlated with MetS; however, this correlation was not found in our results. While our results revealed connections between *APOA5*, *APOC1*, and MetS, the role of *APOE* was not identified. These findings highlight the complex nature of MetS and the need for detailed regional analysis to allow for targeted strategies and global insights.

Similar to our population, several studies within Taiwan have explored the genetic underpinnings of MetS [61,62,63]. One investigation, analyzing nine genes: *APOA5*, *APOC1*, *BRAP*, *BUD13*, *CETP*, *LIPA*, *LPL*, *PLCG1*, and *ZPR1*, examined their association with MetS. This study identified nominal associations between specific gene variants (*APOA5*, *BUD13*, *CETP* and *LIPA*) and MetS, notably linked with high triglyceride and low high-density lipoprotein cholesterol levels. Moreover, it noted gene–environment interactions, such as alcohol consumption and physical activity, influencing the risk of MetS [61]. Another study delved into *APOB* variants, revealing their independent relationships with various lipid levels and MetS traits. Notably, specific *APOB* variants impacted diabetes risk through low-density lipoprotein cholesterol levels [62]. In addition, an exploration of TGF-β pathway genes (*SMAD2* and *TGFBR2*) found significant associations with MetS and its traits, emphasizing gene–gene interactions affecting MetS risk [63]. Lastly, an investigation into circadian genes’ role in MetS used advanced modeling techniques, identifying significant SNPs associated with the syndrome and highlighting potential contributions of circadian gene defects to MetS development [64]. While our findings align with *APOA5*, *BUD13*, and *CETP* associations in most studies, disparities regarding *APOB* variants, *ZPR1*, TGF-β pathway genes and circadian rhythm genes might arise from differing MetS definitions, adjusted covariates, study designs, and methodologies. Overall, these collective findings underscore the intricate genetic contributions and interactions influencing MetS within Taiwanese populations.

In our analysis, we found that glycerolipid metabolism, PPAR signaling, and fatty acid metabolism were pivotal pathways relevant to MetS in the enrolled Taiwanese participants. Glycerolipid metabolism exerts a significant influence on MetS, where dysregulated fatty acid synthesis and triglyceride accumulation underlie key aspects such as obesity, hyperlipidemia, and insulin resistance [65]. Moreover, these metabolic disturbances increase cardiovascular risk through the promotion of fatty acid storage, which is linked to cardiovascular disease [65]. Furthermore, these disturbances can also influence the PPAR signaling pathway, which directly shapes lipid, glucose metabolism, and inflammation dynamics [66,67,68]. The disruption of this pathway has been linked to MetS components including obesity, diabetes mellitus, hypertension, and fatty liver [69]. Targeting PPAR with drugs, particularly fibrates and omega-3 fatty acids, holds therapeutic potential for managing hypertriglyceridemia and blood glucose anomalies [70]. Complementing these pathways, the fatty acid metabolism signaling pathway, which is instrumental for intracellular processes, is also closely connected with MetS. Orchestrating the synthesis, breakdown, and transport of fatty acids, it forms the backbone of lipid metabolism. In the context of MetS, dysregulation of this pathway fosters the synthesis and storage of fatty acids while constraining their oxidation and transport [71,72]. This can then lead to the exacerbation of MetS components including obesity, hypertension, hyperglycemia, and hyperlipidemia.

Our study demonstrated several valuable genetic associations with MetS. It is crucial to discern the mechanistic insights behind how these identified genes contribute to MetS. Figure 4 and Appendix A provide an overview of the mechanisms related to MetS, linking our genetic associations to signaling pathways. *APOA5*, *CETP*, *LPL*, *APOC1*, C19orf80 (*ANGPTL8*), and *MLXIPL* are linked to glycerolipid metabolism, pivotal in lipid metabolism and insulin resistance [73]. *SIK3* and *FOXA2* are associated with the PPAR signaling pathway, influencing lipid and glucose metabolism [74]. *TOMM40*, *GCK*, *FTO*, *VMP1*, *TUBD1*, *SIDT2*, *SENP7*, *PAFAH1B2*, and *DOCK6* relate to fatty acid metabolism [75]. Genes like *ZPR1*, *BUD13*, *YKT6*, *RPS6KB1*, and *BCL7B* do not align directly with these metabolic pathways, potentially serving diverse roles beyond them. It is noteworthy that gene functions and pathway classifications may vary among studies, indicating broader roles beyond defined metabolic pathways.

The limitations of the present study arise from the relatively modest sample size of 107,230 individuals, in comparison to the approximate sample size of 500,000 individuals in the UK Biobank. This discrepancy may curtail the generalizability of our research findings. In addition, 64% of our participants were female, and this sex imbalance could limit the direct applicability of our results to a broader population. Regarding gender-specific associations in MetS, notable genetic findings emerged [47,76,77]. A Tunisian study identified LRPAP1-rs762861 as influential, particularly among females [76]. In a separate Korean study, ZNF664-rs12310367 affected BMI solely in women, while KLF14-rs1562398 correlated with glucose issues in men [47]. Highlighting gender-based vulnerability, specific genetic variants for MetS in females were noted, emphasizing the importance of considering gender-specific genetics in addressing this syndrome [77]. Another limitation is that we lacked cell and animal studies to identify underlying mechanisms. Instead, we relied on openly published data sources such as FUMA and MEGMA. Moreover, we exclusively focused on Taiwanese individuals, which may restrict the applicability of our results to other ethnic groups. However, this population-specific approach can provide more precise insights, elucidating disease characteristics and unique genetic variations within that specific cohort. Finally, different from previous GWASs that separately analyzed individual MetS components, we embraced a holistic approach to MetS. While this approach forgoes the granular detail of associating specific genes with individual disease components, it ensures a concentrated exploration of MetS as an integrated entity. Our rationale was that this comprehensive perspective would yield insights into the overarching mechanisms, risk factors, and potential treatment options for MetS.

## 5. Conclusions

In this study, we identified genomic risk loci for MetS in a large Taiwanese population through a comprehensive GWAS approach. Notably, the *SIK3*, *YKT6*, *RPS6KB1*, and *SENP7* genes have received comparatively less attention in previous research. These associations provide novel insights into the genetic basis of MetS and hold promise for the potential discovery of clinical biomarkers.

## Figures and Tables

**Figure 1 nutrients-16-00077-f001:**
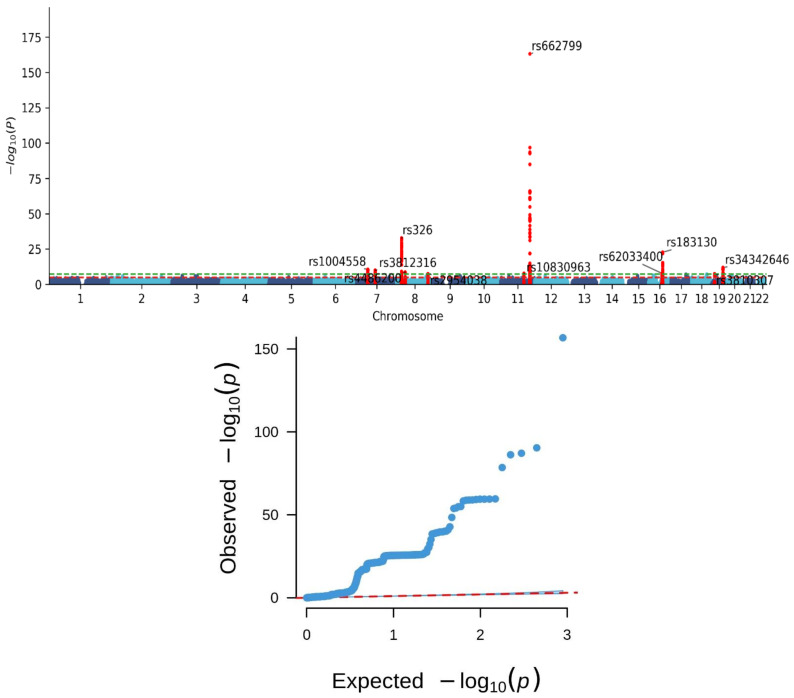
**The Manhattan (upper panel) and Q–Q (lower panel) plots of genome-wide association study for metabolic syndrome.** The Manhattan and Q–Q plots depict the genetic association results of metabolic syndrome phenotypes in a Taiwanese population. The Manhattan plot shows an observed *p*-value (on a -10 log-scale) for each single nucleotide polymorphism in relation to the chromosomal position. The green and red lines in the Manhattan plot represent genome-wide significance with a *p*-value of 5 × 10^−8^ and suggested significance with a *p*-value of 1 × 10^−5^, respectively. The Q-Q plot shows the observed *p*-value versus the expected *p*-value (on a -10 log-scale). The dotted line in the Q-Q plot represents the diagonal line (expected distribution line). Each point on the Q-Q plot corresponds to a specific quantile of the expected distribution (x-axis) and its corresponding observed *p*-value from the analysis (y-axis). Points aligning with the diagonal line indicate that observed *p*-values match the expected distribution, signifying appropriate calibration of the statistical test. Points above the diagonal line represent observed *p*-values that surpass what’s expected by chance under the null hypothesis. This suggests potential true associations or systematic biases in the data or analysis. The Manhattan plot in the present study reveals significant loci clustering on chromosomes 7, 8, 11, 16, and 19. The Q–Q plot exhibits a notable deviation from the expected null distribution, indicating an increased occurrence of lower *p*-values beyond chance alone.

**Figure 2 nutrients-16-00077-f002:**
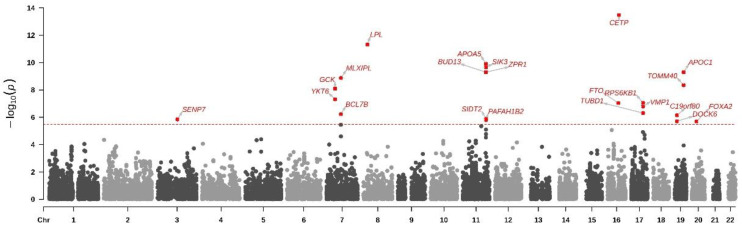
**The Manhattan plot of the gene-based test as computed by MAGMA based on genome-wide association study for metabolic syndrome summary statistics.** The Manhattan plots display the gene-based test computed by MAGMA using summary statistics from genome-wide association study for metabolic syndrome in a Taiwanese population. The color dots show an observed *p*-value (on a -10 log-scale) for each gene in relation to the chromosomal position. The line on the plot represents gene-based significance with a *p*-value of 3.27 × 10^−6^.

**Figure 3 nutrients-16-00077-f003:**
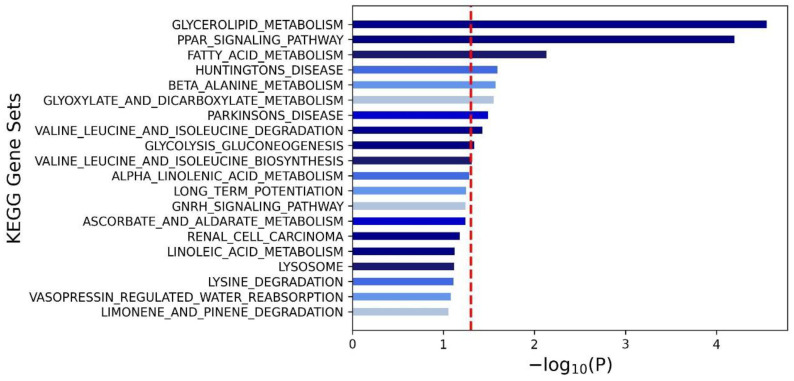
Significant pathways (with a false discovery rate (FDR) of <0.05) that were associated with metabolic syndrome. The red dotted line represents that the threshold of the *p*-value after the Bonferroni correction.

**Figure 4 nutrients-16-00077-f004:**
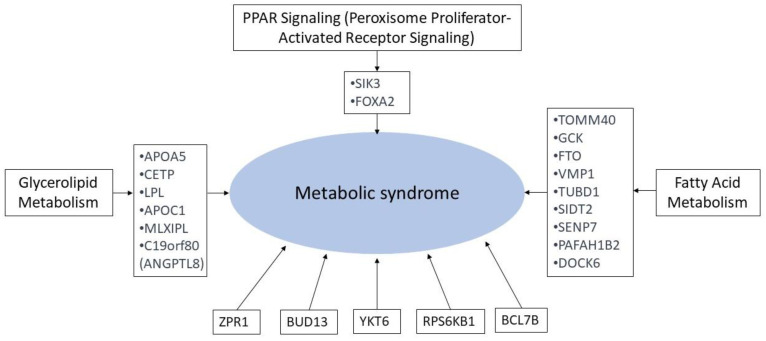
An overview of the mechanisms related to MetS, linking our genetic associations to signal pathways.

**Table 1 nutrients-16-00077-t001:** Clinical characteristics of the study participants.

Characteristics	All	Metabolic Syndrome	No Metabolic Syndrome
n	107,230	24,171	83,059
**Demographic data**			
Age, yrs	49.91 ± 10.92	53.99 ± 10.1	48.83 ± 10.92
Women, n (%)	68,641 (64)	13,567 (56)	55,074 (66)
BMI, kg/m^2^	24.22 ± 3.77	27.27 ± 3.76	23.33 ± 3.28
Systolic BP, mm Hg	120.28 ± 18.6	133.03 ± 17.9	116.57 ± 17.1
Diastolic BP, mm Hg	73.74 ± 11.35	80.63 ± 11.27	71.74 ± 10.57
**Metabolic syndrome and its components**			
Metabolic syndrome, n (%)	24,171 (23)	24,171 (100)	0 (0)
Hypertension *, n (%)	37,603 (35)	18,060 (75)	19,543 (24)
Impaired glucose tolerance †, n (%)	22,131 (21)	13,522 (56)	8609 (10)
Increased waist circumference ‡, n (%)	49,870 (47)	21,178 (88)	28,692 (35)
Hypertriglyceridemia §, n (%)	22,397 (21)	15,463 (64)	6934 (8)
Low high-density lipoprotein ǁ, n (%)	27,677 (26)	15,911 (66)	11,766 (14)

BMI = Body mass index; BP = Blood pressure; All variables showed significant differences (*p*-value < 0.05) between subjects with and without metabolic syndrome; * Systolic blood pressure greater than 130 mm Hg or diastolic blood pressure greater than 85 mm Hg; † Fasting glucose level greater than 100 mg/dL; ‡ Waist circumference greater than 90 cm in men and greater than 80 cm in women; § Serum triglyceride level greater than 150 mg/dL; ǁ High-density lipoprotein cholesterol level less than 40 mg/dL.

**Table 2 nutrients-16-00077-t002:** A summary of genomic risk loci identified in the genome-wide association analysis for metabolic syndrome.

SNP	Chr	Position	Effect Allele	Other Allele	EAF	Beta Coefficient	SE	*p*	Nearest Gene(s)
rs1004558	7	44240407	C	T	0.21	0.09	0.01	2.78 × 10^−11^	*YKT6*
rs3812316	7	73020337	C	G	0.08	−0.12	0.02	1.06 × 10^−10^	*MLXIPL*
rs326	8	19819439	A	G	0.19	−0.16	0.01	1.71 × 10^−33^	*LPL*
rs4486200	8	34293752	C	T	0.50	0.06	0.01	2.31 × 10^−9^	*RPL10AP3*, *LINC01288*
rs2954038	8	126507389	A	C	0.29	0.06	0.01	3.33 × 10^−8^	*TRIB1*, *LINC00861*
rs10830963	11	92708710	C	G	0.44	0.06	0.01	1.80 × 10^−8^	*MTNR1B*
rs662799	11	116663707	A	G	0.32	0.31	0.01	7.32 × 10^−164^	*APOA5*
rs62033400	16	53811788	A	G	0.13	0.09	0.02	1.52 × 10^−8^	*FTO*
rs183130	16	56991363	C	T	0.15	−0.15	0.01	1.86 × 10^−23^	*CETP*, *HERPUD1*
rs34342646	19	45388130	G	A	0.09	0.13	0.02	1.41 × 10^−12^	*NECTIN2*

SNP = single nucleotide polymorphism; Chr = chromosome; EAF = effect allele frequency; SE: standard error of the beta coefficient.

**Table 3 nutrients-16-00077-t003:** Lists of key genes associated with metabolic syndrome in the study population.

Gene	Chr	NSNPS	Z	*p*
*CETP*	16	42	7.50	3.25 × 10^−14^
*LPL*	8	31	6.81	4.77 × 10^−12^
*APOA5*	11	4	6.33	1.22 × 10^−10^
*SIK3*	11	473	6.24	2.21 × 10^−10^
*ZPR1*	11	14	6.11	5.00 × 10^−10^
*APOC1*	19	2	6.11	5.00 × 10^−10^
*BUD13*	11	15	6.11	5.00 × 10^−10^
*MLXIPL*	7	19	5.95	1.36 × 10^−9^
*TOMM40*	19	10	5.75	4.45 × 10^−9^
*GCK*	7	49	5.65	7.99 × 10^−9^
*YKT6*	7	24	5.34	4.71 × 10^−8^
*RPS6KB1*	17	60	5.22	8.74 × 10^−8^
*FTO*	16	339	5.21	9.26 × 10^−8^
*VMP1*	17	104	5.11	1.62 × 10^−7^
*TUBD1*	17	37	4.90	4.88 × 10^−7^
*BCL7B*	7	2	4.86	6.00 × 10^−7^
*C19orf80 (ANGPTL8)*	19	1	4.82	7.14 × 10^−7^
*SIDT2*	11	24	4.69	1.34 × 10^−6^
*SENP7*	3	378	4.68	1.42 × 10^−6^
*PAFAH1B2*	11	60	4.66	1.57 × 10^−6^
*DOCK6*	19	36	4.62	1.95 × 10^−6^
*FOXA2*	20	4	4.61	2.04 × 10^−6^

Chr = chromosome; NSNPS = the number of single nucleotide polymorphisms associated with the gene.

## Data Availability

The data underlying this study is from the Taiwan Biobank. Due to restrictions placed on the data by the Personal Information Protection Act of Taiwan, the minimal data set cannot be made publicly available. Data may be available upon request to interested researchers. Please send data requests to: Szu-Chia Chen, Division of Nephrology, Department of Internal Medicine, Kaohsiung Medical University Hospital, Kaohsiung Medical University.

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
