# Peer review of "A Genome-Wide Association Study of Metabolic Syndrome in the Taiwanese Population"

_nutrients, 2023, doi:10.3390/nu16010077_

Round 1

Reviewer 1 Report

Comments and Suggestions for Authors

The manuscript entitled "A genome-wide association study (GWAS) of metabolic syndrome in the Taiwanese population" examines the genetic basis of metabolic syndrome in the Taiwanese population using a GWAS approach. The article is well-organized, and the methodology is described in detail. The study fills a gap in MetS research, particularly in Asian populations, and presents new findings on genetic associations with MetS.

The introduction provides a detailed background to MetS, its global prevalence, associated risk factors, and the existing research gap, thus laying the foundation for the study.

The methods section is detailed and well-structured and clearly explains the data sources, study population, genotyping, quality control procedures, and statistical analyses.

The results section is comprehensive and presents the clinical characteristics of the study participants, the GWAS analysis, the SNPs identified, the genes associated with MetS, and the signaling pathways involved. The inclusion of figures (Manhattan plots, Q-Q plots) increases the clarity of the results.

The identification of new genomic risk loci, including the less-studied genes SIK3, YKT6, RPS6KB1, and SENP7, adds a new dimension to the study and contributes to the understanding of the genetic basis of MetS.

The article effectively compares its findings with previous studies, highlighting similarities and differences. This contextualization underscores the importance of the study's contributions.

Areas for improvement:

Authors should avoid the use of abbreviations in the title (GWAS).

The article notes a clear gender imbalance among the study participants: 64% were women. The implications of this imbalance for the generalizability of the study and whether gender-specific associations were explored should be discussed.

While the study provides valuable genetic associations, mechanistic insights into how the identified genes may contribute to MetS are still lacking. Discussion of possible mechanisms or suggestions for future mechanistic studies would enhance the article.

Overall, the article is a valuable contribution to understanding the genetic basis of MetS in the Taiwanese population. Addressing the mentioned areas for improvement further enhances the clarity and impact of the study.

Author Response

Dear editor and reviewers:

We are very grateful to your comments for the manuscript. We also appreciate the time and effort you and each of the reviewers have dedicated to providing insightful feedback on ways to strengthen our paper. Thus, it is with great pleasure that we resubmit our article for further consideration. We have incorporated changes that reflect the detailed suggestions you have graciously provided. We also hope that our edits and the responses we provide below satisfactorily address all the issues and concerns you and the reviewers have noted.

To facilitate your review of our revisions, the following is a point-by-point response to the questions and comments delivered in your letter dated 11/30/2023.

Reviewer 1 Comments:

  1. Authors should avoid the use of abbreviations in the title (GWAS).
  • Response: Thank you for your suggestion, we have revised the title as “A genome-wide association study of metabolic syndrome in the Taiwanese population.”
  1. The article notes a clear gender imbalance among the study participants: 64% were women. The implications of this imbalance for the generalizability of the study and whether gender-specific associations were explored should be discussed.
  • Response: We wholeheartedly agree with the perspective that gender is an important factor associated with metabolic syndrome. In our present study, an imbalanced gender distribution stands as one of our limitations, as described in our manuscript: “In addition, 64% of our participants were female, and this sex imbalance could limit the direct applicability of our results to a broader population.” Furthermore, in our GWAS analysis, we accounted for gender as a factor to investigate candidate SNPs linked to metabolic syndrome independent of gender influences. Moreover, we included a discussion on gender-specific associations within metabolic syndrome.
  • Please find the revised version “ In addition, 64% of our participants were female, and this sex imbalance could limit the direct applicability of our results to a broader population. Regarding gender-specific associations in MetS, notable genetic findings emerged [41,77,78]. The Tunisian study identified LRPAP1-rs762861 as influential, particularly among females [77]. In a separate Korean study, ZNF664-rs12310367 affected BMI solely in women, while KLF14-rs1562398 correlated with glucose issues in men [41]. Highlighting gender-based vulnerability, specific genetic variants for MetS in females were noted, emphasizing the importance of considering gender-specific genetics in addressing this syndrome [78]. 
  1. While the study provides valuable genetic associations, mechanistic insights into how the identified genes may contribute to MetS are still lacking. Discussion of possible mechanisms or suggestions for future mechanistic studies would enhance the article.
  • Response: No problem. In the discussion section, we extensively covered several SNPs/Genes, including: 1. CETP, which facilitates the transfer of cholesterol esters and triglycerides between lipoproteins, impacting cholesterol metabolism and lipid balance significantly [30, 39]. 2. LPL, crucial in lipoprotein metabolism and the utilization of fatty acids, hydrolyzes triglycerides in tissues, releasing energy-rich fatty acids [38]. 3. APOA5, playing a pivotal role in lipid metabolism [37]. 4. SIK3, encoding a protein involved in regulating glucose, lipid metabolism, and neuronal development, with potential implications in obesity, metabolic disorders, and neurodegenerative diseases [34, 47]. 5. BUD13, ZPR1, and APOA5, implicated in MetS-related events, notably correlating with elevated serum triglyceride levels [43]. 6. APOC1, APOA5, playing crucial roles in lipid metabolism [37]. 7. MLXIPL, associated with glycerolipid Metabolism. 8. TOMM40, linked to fatty acid metabolism. 9. GCK, crucial in glucose phosphorylation, and associated with diabetes and metabolic disorders [44-46]. 10. YKT6, associated with CD8+ T cell levels and under exploration as a potential biomarker for oral squamous cell carcinoma, its relevance to MetS is under investigation [48-50]. 11. RPS6KB1, a kinase linked to various cellular processes and protein synthesis, showing associations with conditions like fatty liver disease, intestinal disorders, and oxidative stress pathways [50-52]. 12. SENP7, a regulator of SUMOylation, playing roles in maintaining CD8+ T cell metabolic status and contributing to muscle sarcomere organization [53, 54]. 13. FTO, VMP1, TUBD1, SIDT2, SENP7, PAFAH1B2, DOCK6, all related to fatty acid metabolism. 14. BCL7B, involved in cell cycle regulation and transcriptional control; its mechanism in MetS requires further study. 15. C19orf80 (ANGPTL8), involved in regulating lipoprotein lipase crucial for lipid metabolism, linked to the glycerolipid metabolism pathway. 16. FOXA2, related to PPAR Signaling pathways.
  • We recognize that the complexities of these SNPs/Genes and their mechanisms might confuse our readers. Therefore, we've prepared a table and a figure to illustrate these relationships. Please find supplementary Table 2 and Figure 4 in the revised manuscript.
  • We have included an additional paragraph to provide an overview of these genes as follows: “Our study demonstrated several valuable genetic associations with MetS. It’s cru-cial to discern the mechanistic insights behind how these identified genes contribute to MetS. Figure 4 and Table S2 provide an overview of the mechanisms related to MetS, linking our genetic associations to signal pathways. APOA5, CETP, LPL, APOC1, C19orf80 (ANGPTL8), and MLXIPL are linked to glycerolipid metabolism, pivotal in lipid metabolism and insulin resistance [74]. SIK3 and FOXA2 are associated with the PPAR signaling pathway, influencing lipid and glucose metabolism [75]. TOMM40, GCK, FTO, VMP1, TUBD1, SIDT2, SENP7, PAFAH1B2, and DOCK6 relate to fatty acid metabolism [76]. Genes like ZPR1, BUD13, YKT6, RPS6KB1, and BCL7B don't align directly with these pathways, potentially serving diverse roles beyond specified met-abolic pathways. It's noteworthy that gene functions and pathway classifications may vary among studies, indicating broader roles beyond defined metabolic pathways.” (line 445-456)

Reviewer 2 Report

Comments and Suggestions for Authors

Ho and colleagues investigated the genetic factors associated with metabolic syndrome (MetS) in a sample population from Taiwan in a genome-wide association study (GWAS). They analysed comprehensive clinical and genetic information from 107,230 Taiwanese individuals using more than 650,000 single nucleotide polymorphisms (SNPs). MetS was defined using a modified version of the ATPIII criteria. 549 SNPs were identified that were significantly associated with MetS and collectively linked to 10 genomic risk loci. Ten SNPs (rs1004558, rs3812316, rs326, rs4486200, rs2954038, rs10830963, rs662799, rs62033400, rs183130 and rs34342646) and 22 associated genes were identified.

Comments and suggestions:

-          The introduction is short and does not include information on the situation of metabolic syndrome in Taiwan. However, it does present data from the US that do not apply to populations of Asian origin. It also does not give the reader sufficient information on the genetic determinants of metabolic syndrome. I recommend a revision.

-          I recommend the consistent use of decimal points throughout the entire manuscript. Sometimes authors present results to 10 decimal places and sometimes to 2.

-          In the first table, it would be useful to compare the metabolic syndrome and METs-free groups in addition to the total population.

-          I suggest a flowchart showing the sampling of the sample population.

-          The discussion lacks relevant publications on the subject. I recommend a thorough review of the literature and a supplement to the discussion.

o   Lin E, Kuo PH, Liu YL, Yang AC, Kao CF, Tsai SJ. Association and interaction of APOA5, BUD13, CETP, LIPA and health-related behaviour with metabolic syndrome in a Taiwanese population. Sci Rep. 2016 Nov 9;6:36830. doi: 10.1038/srep36830. PMID: 27827461; PMCID: PMC5101796.

o   Jang, S.-J.; Tuan, W.-L.; Hsu, L.-A.; Er, L.-K.; Teng, M.-S.; Wu, S.; Ko, Y.-L. Pleiotropic Effects of APOB Variants on Lipid Profiles, Metabolic Syndrome, and the Risk of Diabetes Mellitus. Int. J. Mol. Sci. 2022, 23, 14963. https://doi.org/10.3390/ijms232314963

o   Lin, E., Kuo, PH., Liu, YL. et al. Transforming growth factor-β signalling pathway-associated genes SMAD2 and TGFBR2 are implicated in metabolic syndrome in a Taiwanese population. Sci Rep 7, 13589 (2017). https://doi.org/10.1038/s41598-017-14025-4

o   Hsu NW, Chou KC, Wang YT, Hung CL, Kuo CF, Tsai SY. Building a model for predicting metabolic syndrome using artificial intelligence based on an investigation of whole-genome sequencing. J Transl Med. 2022 Apr 28;20(1):190. doi: 10.1186/s12967-022-03379-7. PMID: 35484552; PMCID: PMC9052619.

Author Response

Dear editor and reviewers:

We are very grateful to your comments for the manuscript. We also appreciate the time and effort you and each of the reviewers have dedicated to providing insightful feedback on ways to strengthen our paper. Thus, it is with great pleasure that we resubmit our article for further consideration. We have incorporated changes that reflect the detailed suggestions you have graciously provided. We also hope that our edits and the responses we provide below satisfactorily address all the issues and concerns you and the reviewers have noted.

To facilitate your review of our revisions, the following is a point-by-point response to the questions and comments delivered in your letter dated 11/30/2023.

Reviewer 2 Comments:

This is a well-written and easy to understand study of a topic that has not been much investigated. The results are nicely presented in tables. However, I have concerns about the methods used, which are a somewhat outdated and not well described. I summarize my comments below.

Introduction/background:

  1. The introduction is short and does not include information on the situation of metabolic syndrome in Taiwan. However, it does present data from the US that do not apply to populations of Asian origin. It also does not give the reader sufficient information on the genetic determinants of metabolic syndrome. I recommend a revision.
  • Response: Thank you for your suggestion. We have revised the introduction to include information about metabolic syndrome in Taiwan and the genetic determinants associated with it. Please find the revised manuscript: “In Taiwan, there has been a notable increase in the prevalence of MetS, affecting 9.5% of the population based on the strict NCEP III standards. In addition, the prevalence rises to 12.9% when considering the Asian waist circumference criteria [7].” (line 55-56) and “The genetic factors contributing to MetS encompass intricate interactions among multiple genes and environmental influences [8]. While lifestyle choices, notably diet and exercise, wield significant impact, genetic predispositions play a substantial role in determining an individual's susceptibility to this condition [9]. Several studies have identified numerous genes linked to lipid metabolism, insulin resistance, inflammation, and obesity, with specific variants such as LPL, CETP, and APOA5 demonstrating heightened susceptibility [10-12]. The heritability of MetS, observable in families with a history of diabetes, hypertension, or obesity, underscores its genetic foundation [13]. Understanding the genetic basis of MetS is critical for tailored medical approaches, yet the condition's polygenic nature complicates the identification of individual genes. Large-scale studies aimed at identifying genetic markers offer promise in unraveling its complexity.” (line 61-71)
  1. I recommend the consistent use of decimal points throughout the entire manuscript. Sometimes authors present results to 10 decimal places and sometimes to 2.
  • Response: No problem, we round the decimals to two places. Please find the revised versions of Table 1, Table 2, and Table 3.
  1. In the first table, it would be useful to compare the metabolic syndrome and METs-free groups in addition to the total population.
  • Response: We have revised Table 1 to include information about both the metabolic syndrome and MetS-free groups. In addition, we've made revisions in the 'Materials and Methods' and 'Results' sections to elucidate how we delineate the variables between these two groups and expound upon the differences observed.
  • Please find the revised versions of Table 1.
  • Please find the revised manuscript in the 'Materials and Methods': “The clinical characteristics of subjects, represented as mean ± standard deviation (SD) for continuous variables and as n (%) for categorical variables, underwent comparison using the chi-square test for categories and the independent t-test for continuous traits, considering a P-value < 0.05 as indicative of significant differences.” (line 205-208)
  • Please find the revised manuscript in the 'Results': “A comparison between subjects with and without MetS showed that those with MetS were older, had higher BMI, and exhibited elevated blood pressure levels (Table 1).”(227-229)
  1. I suggest a flowchart showing the sampling of the sample population.
  • Response: Not a problem, we've included Supplementary Figure 1 depicting the flowchart.
  1. The discussion lacks relevant publications on the subject. I recommend a thorough review of the literature and a supplement to the discussion.
  • Response: Thank you for your suggestion. We have reviewed these articles and made revisions to our manuscript as follows: “Similar to our population, several studies within Taiwan have explored the genetic underpinnings of MetS [42-45]. One investigation, analyzing nine genes: APOA5, APOC1, BRAP, BUD13, CETP, LIPA, LPL, PLCG1, and ZPR1, examined their associa-tion with MetS. This study identified nominal associations between specific gene var-iants (APOA5, BUD13, CETP and LIPA) and MetS, notably linked with high triglyc-eride and low high-density lipoprotein cholesterol levels. Moreover, it noted gene-environment interactions, such as alcohol consumption and physical activity, in-fluencing the risk of MetS [42]. Another study delved into APOB variants, revealing their independent relationships with various lipid levels and MetS traits. Notably, specific APOB variants impacted diabetes risk through low-density lipoprotein cho-lesterol levels [43]. In addition, an exploration of of TGF-β pathway genes (SMAD2 and TGFBR2) found significant associations with MetS and its traits, emphasizing gene-gene interactions affecting MetS risk [44]. Lastly, an investigation into circadian genes' role in MetS used advanced modeling techniques, identifying significant SNPs associated with the syndrome and highlighting potential contributions of circadian gene defects to MetS development [45]. While our findings align with APOA5, BUD13, and CETP are associations in most studies, disparities regarding APOB variants, TGF-β pathway genes, and circadian rhythm genes might arise from differing MetS defini-tions, adjusted covariates, study designs, and methodologies. Overall, these collective findings underscore the intricate genetic contributions and interactions influencing MetS within Taiwanese populations.” (line 367-386)

CONCLUDING REMARKS: Again, thank you for giving us the opportunity to strengthen our manuscript with your valuable comments and queries. We have worked hard to incorporate your feedback and hope that these revisions persuade you to accept our submission.

Sincerely,

JIUN-HUNG GENG

Round 2

Reviewer 2 Report

Comments and Suggestions for Authors

I accept the authors' answers to my questions and comments.

Author Response

Thank you for giving us the opportunity to strengthen our manuscript with your valuable comments and queries.